# Substituted cysteine modification and protection indicates selective interactions of the anesthetic photolabel pTFD-di-iPr-BnOH with α+/β– and α+/γ– transmembrane subunit interfaces of synaptic GABA<sub>A</sub> receptors

**Kieran Bhave, Stuart A. Forman**[ID]*

Department of Anesthesia Critical Care & Pain Medicine, Massachusetts General Hospital, Boston, Massachusetts, United States of America

* saforman@mgh.harvard.edu

## Abstract

### Background

General anesthesia induced by etomidate, barbiturates and propofol is associated with positive modulation of synaptic αβγ GABA<sub>A</sub> receptors, inhibitory hetero-pentameric ligand-gated ion channels formed from homologous subunits arranged β-α-β-α-γ around a central gated chloride channel. Approaches based on mutations, amino-acid level analysis of photolabel incorporation, and cryo-electron micrography (cryo-EM) all indicate that etomidate binds selectively in two outer transmembrane β+/α– inter-subunit sites per receptor. These approaches also reveal that the potent barbiturate photolabel R-mTFD-MPAB binds selectively in homologous sites formed at α+/β– and γ+/β– interfaces. The anesthetic photolabel, pTFD-di-iPr-BnOH, was proposed to bind selectively in α+/β– and α+/γ– homologs of the etomidate sites, based largely on functional analysis of only 5 point mutations in α1β3γ2L receptors.

### Methods

To further test the interactions of receptor-bound pTFD-di-iPr-BnOH with outer transmembrane inter-subunit sites, we used voltage-clamp electrophysiology in substituted cysteine modification and protection (SCAMP) experiments at 8 residues located in the five homologous sites, focusing on α+ and γ– loci. Control SCAMP studies were performed using etomidate and R-mTFD-MPAB.

### Results

Incorporation of single cysteine mutations (α1M236C, α1S280C, α1A291C, β3L231C, β3M286C, γ2I242C, γ2L246C, and γ2S301C) produced functional GABA-responsive

**Data availability statement:** All relevant data are within the paper and its Supporting Information files.

**Funding:** SAF received supported for this work from a grant from the US National Institutes of Health (R35GM141951). The funders had no role in study design, data collection and analysis, decision to publish or preparation of the manuscript.

**Competing interests:** The authors have declared that no competing interests exist.

receptors that retained sensitivity to pTFD-di-iPr-BnOH modulation and displayed increased GABA sensitivity following exposure to the covalent sulfhydryl modifier p-chloromercuribenzenesulfonate (pCMBS). In the presence of pTFD-di-iPr-BnOH, pCMBS modification effects were reduced (evidence of steric protection) in receptors with cysteine mutations in α+, β–, and γ–, but not in α–, β+, or γ+ interfacial loci. Protection patterns with etomidate and R-mTFD-MPAB mirrored prior results.

## Discussion

SCAMP results further support the hypothesis that pTFD-di-iPr-BnOH binds selectively in α+/β– and α+/γ– interfacial sites that are homologs of the β+/α– etomidate sites.

## Introduction

Gamma-aminobutyric acid type A (GABA$_A$) receptors are pentameric ligand-gated chloride channels and the major inhibitory neurotransmitter receptors in the mammalian central nervous system. Typical synaptic GABA$_A$ receptors contain 2α, 2β, and 1γ subunit pseudo-symmetrically arranged around the central gated chloride channel (β-α-β-α-γ anticlockwise viewed from an extracellular perspective). Each homologous subunit contains four transmembrane alpha helices (M1 through M4) with M2 helices forming the ion channel. Subunit structures abutting adjacent subunits are designated as "+" (M3) or "–" (M1) [1]. Different aspects of M2 helices also contribute to + and – interfaces.

Clinical drugs that positively modulate GABA$_A$ receptors include the intravenous general anesthetics etomidate, propofol, barbiturates, and the neurosteroid alphaxalone. Molecular sites mediating modulation of GABA$_A$ receptors by these drugs have been explored using various approaches including photolabeling with proteomic analyses [2], cryo-electron micrography [3], x-ray crystallography [4], electrophysiologic mutant-function studies [5–7], and substituted cysteine modification with drug protection (SCAMP), also based on electrophysiology [8,9]. Each of these methods presents different advantages and disadvantages. Robust models of anesthetic-receptor interactions are based on convergent results from multiple approaches.

The most thoroughly established GABA$_A$ receptor-anesthetic interactions are for R-etomidate and the potent stereoselective barbiturate photolabel R-5-allyl-1-methyl-5-(m-trifluoromethyl-diazirynylphenyl) barbituric acid (R-mTFD-MPAB). Early mutant-function studies in heterologously expressed multimeric receptors identified β2/3N265 and β2/3M286 as determinants of etomidate sensitivity [10]. The photoreactive etomidate analog azi-etomidate labeled α1M236 and β2/3M286 in receptors purified from bovine brain, suggesting that etomidate binds in two transmembrane β+/α– interfacial sites per receptor [11]. Additional mutant-function, SCAMP, and cryo-EM studies [3,7,9,12,13] confirmed that etomidate binds near α1L232, α1M236 (both in M1), β2/3N265 (in M2) and β2/3M286 (in M3). The barbiturate photolabel R-mTFD-MPAB incorporates in α1β3γ2 receptors at β3M227 (in

M1), α1A291 and α1Y294 (both in M3) as well as γ2S301 (in M3) [14], indicating selectivity for α+/β– and γ+/β– interfacial sites that are homologs of etomidate β+/α– sites. Mutant-function and SCAMP studies [9] confirmed these findings and cryo-EM imaging [3] locates phenobarbital near these residues. Thus, of the 5 transmembrane intersubunit interfaces per αβγ receptor, etomidate binds to the two β+ interfaces and R-mTFD-MPAB binds to the two β– interfaces. Fig 1 depicts the arrangement of subunits and transmembrane helices in α1β3γ2L receptors and the loci where etomidate (red diamonds) and R-mTFD-MPAB (green rectangles) selectively bind in homologous transmembrane interfacial sites.

Another anesthetic photolabel, 2,6-diisopropyl-4–3-(trifluormethyl)-diazirin-3-yl)phenyl)methanol (pTFD-di-iPr-BnOH), acts through sites in α1β3γ2L receptors that remain tentative because photoincorporation of the radiolabeled compound has not been characterized at the amino acid level [15]. Subunit-level photoincorporation studies suggested that pTFD-di-iPr-BnOH competes for occupation of one of the two R-mTFD-MPAB sites, but not etomidate sites. Functional analysis of 5 mutations in α1β3γ2L receptors suggested that pTFD-di-iPr-BnOH binds within α+/β– and α+/γ– interfacial pockets.

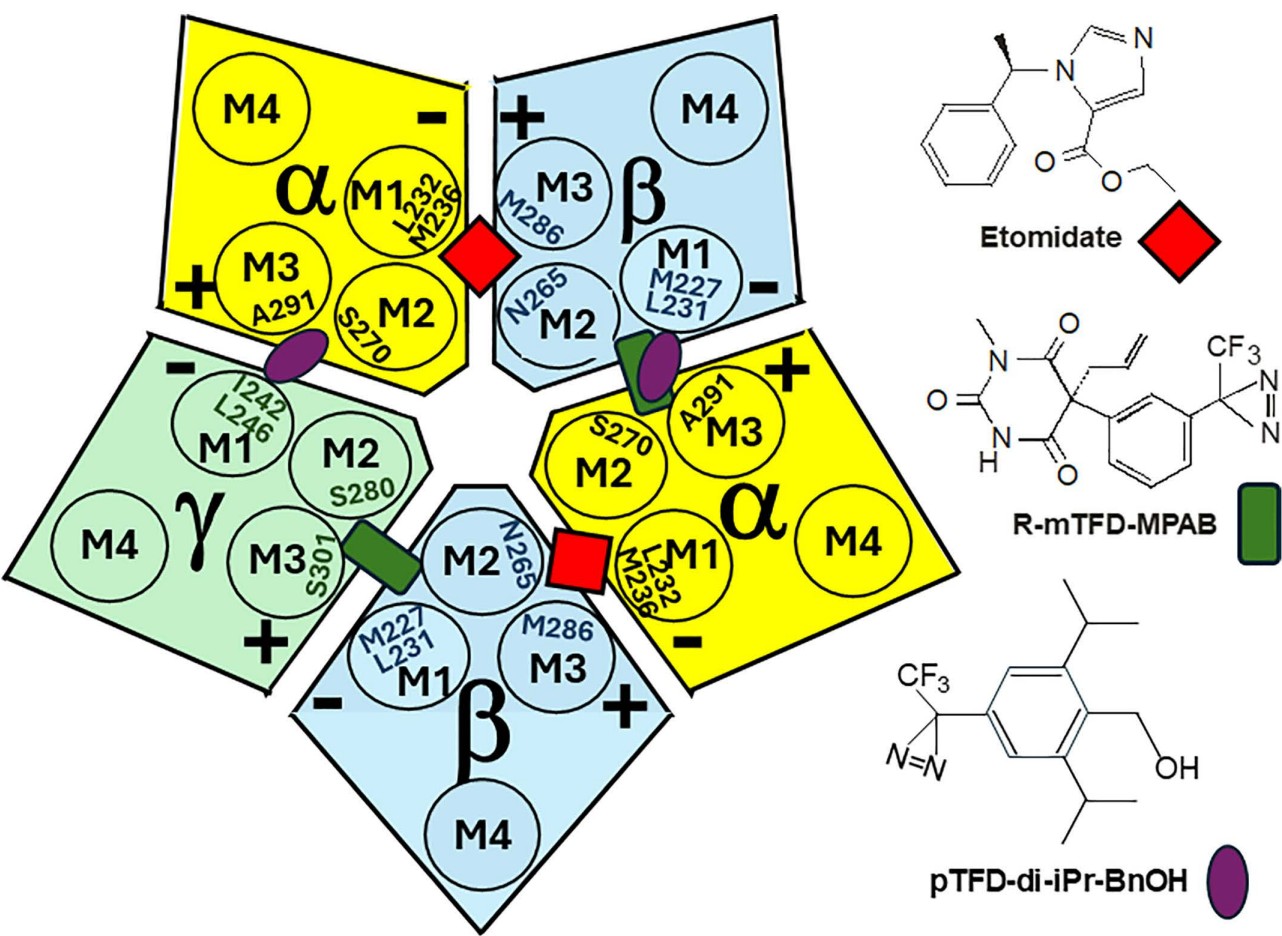

**Fig 1. Transmembrane Inter-subunit Anesthetic Binding Pockets in GABAₐ Receptors.** Left: The diagram depicts the subunit arrangement of α1β3γ2L GABAₐ receptors near the extracellular end of the transmembrane domains. Subunits are color coded: α = yellow; β = blue; and γ = green. The transmembrane helices (M1-M4) of each subunit are labeled, as are the "+" and "–" faces of adjacent subunits. Also depicted are the homologous interfacial pockets where etomidate (red diamonds), R-mTFD-MPAB (dark green rectangles), and pTFD-di-iPr-BnOH (purple ovals) are thought to bind selectively. Twelve amino acid residues that abut these homologous anesthetic binding sites are labeled on corresponding transmembrane helices. These amino acids are positioned at varying depths along the extracellular to intracellular axis. Right: Molecular structures of the three site-selective anesthetics used in these experiments.

The α+/β– interface is one of the two m-TFD-MPAB binding sites, while the homologous α+/γ– transmembrane interface was designated an "orphan" anesthetic site with no well-established drug interactions [9]. The proposed homologous sites where pTFD-di-iPr-BnOH (purple ovals) selectively binds are also shown in Fig 1.

Our current study aimed to probe the interactions of pTFD-di-iPr-BnOH with all 5 GABA$_A$ receptor subunit interfaces using SCAMP, which correlates with photolabeling better than mutant-function tests and is likely less sensitive to allosteric interactions [9]. We used mutated α1β3γ2L receptors with single cysteine substitutions at homologous + and – transmembrane interfacial residues of α1, β3, and γ2L. For each subunit, we selected at least one + and one – mutation, mostly from those used in our previous studies of etomidate and R-mTFD-MPAB. Because pTFD-di-iPr-BnOH interactions at the α+/γ– transmembrane interface were of particular interest, we studied two mutations on each side of this pocket. Thus, eight of the 12 labeled residues depicted in Fig 1 were studied. Using voltage-clamp electrophysiology, we characterized each mutant receptor for its GABA sensitivity, modulation by pTFD-di-iPr-BnOH, and the effects of a sulfhydryl modifying reagent after varying exposures. We then tested whether covalent modification of each substituted cysteine was inhibited in the presence of pTFD-di-iPr-BnOH, using etomidate and R-mTFD-MPAB as positive and negative SCAMP controls. For the two γ– mutants, we used etomidate as the sole negative control, because its binding sites are best established by multiple methods. The protection patterns for the three drugs were compared to previous SCAMP and mutant function results and analyzed for interfacial selectivity patterns.

## Materials and methods

### Animals

Female *Xenopus laevis* were used as sources for oocyte collection in accordance with the Guide for the Care and Use of Laboratory Animals of the National Institutes of Health. Approval for animal use was granted by the Massachusetts General Hospital Institutional Animal Care and Use Committee (protocol #2005N000051). Frogs were housed and maintained in a veterinarian-supervised facility in line with ARRIVE guidelines. Oocytes were harvested *via* mini-laparotomy procedures from frogs anesthetized in 0.2% tricaine. All efforts were made to minimize the suffering of frogs.

### Materials

The sulfhydryl modifying reagent para-chloromercuribenzenesulfonic acid sodium salt (pCMBS) was purchased from Toronto Research Chemicals (Toronto, ON, CA) and stored as a powder at −20°C. Fresh pCMBS stock solutions were prepared daily and kept on ice before dilution in electrophysiology buffer for experiments. R-etomidate (ETO), synthesized by Bachem America (Torrance, CA), was a gift from Prof. Douglas Raines (Department of Anesthesia Critical Care & Pain Medicine, Massachusetts General Hospital Boston, MA, USA) and stored as a 100 mM stock in DMSO at −20°C. R-5-allyl-1-methyl-5-(m-trifluoromethyl-diazirynylphenyl) barbituric acid (R-mTFD-MPAB) and 2,6-diisopropyl-4–3-(trifluormethyl)-diazirin-3-yl)phenyl)methanol (p-TFD-di-iPr-BnOH) were gifts from Prof. Karol Bruzik, PhD (Department of Medicinal Chemistry and Pharmacgnosy, University of Illinois, Chicago, USA) and Prof. Keith W. Miller (Department of Anesthesia Critical Care & Pain Medicine, Massachusetts General Hospital Boston, MA, USA). R-mTFD-MPAB was stored as a 100 mM solution in DMSO and p-TFD-di-iPr-BnOH was stored as a 10mM solution in DMSO. Stock solutions were diluted in electrophysiology buffer for experiments. γ-Aminobutyric acid (GABA), buffers, salts, and antibiotics were all purchased from Sigma-Aldrich (St. Louis, MO, USA).

### Molecular biology

Complementary DNAs encoding human α1, β3, and γ2L in pCDNA3.1 expression vectors were used. Cysteine mutations at the amino acids of interest have been previously described [9,12,16] or were introduced into subunit plasmids by site-directed mutagenesis using QuikChange kits (Agilent Technologies, Santa Clara, CA, USA) and confirmed by sequencing. The mutations were α1M236C, α1S270C, α1A291C, β3L231C, β3M286C, γ2S301C, γ2I242C, and γ2L246C. Capped messenger RNAs were synthesized on linearized DNA templates using mMessage mMachine kits (Thermo

Fisher, Waltham, MA, USA) and, after purification, were mixed in ratios of 1α:1β:5γ [17]. Messenger RNA mixes were diluted to 1ng/nl in nuclease-free water and stored at −80 °C.

## Oocyte expression of GABA$_A$ receptors

Xenopus oocytes were harvested from anesthetized frogs and defolliculated with collagenase digestion as previously described [18]. Defolliculated oocytes were injected with ~50 ng messenger RNA mixtures. Oocytes were incubated at 18 °C for 16–48 hours in ND-96 buffer solution (96 mM NaCl, 3 mM KCL, 1.8 mM MgCl$_2$, 5 mM Hepes, pH 7.4) supplemented with 0.05 mg/ml gentamicin, 0.1 mg/ml ampicillin, and 0.025 mg/ml ciprofloxacin.

## Two electrode voltage-clamp electrophysiology

Experiments were performed in ND-96 buffer at room temperature (20–22°C). Oocytes were placed in a low-volume (30µl) flow chamber and impaled with borosilicate microelectrodes filled with 3M KCl (resistance 0.5–3 MΩ) and voltage-clamped at −50 mV (model OC-725C; Warner Instruments, Hamden, CT, USA). Superfusate solutions in ND-96 were delivered at 2 ml/min from glass syringe reservoirs via computer-controlled valves (VC-8; Warner Instruments), a micro-manifold (ALA-VM8; ALA Scientific Instruments, Farmingdale, NY, USA), and PTFE tubing. Currents were digitized at 200 Hz on a computer running ClampEx v8.0 software (Molecular Devices, San Jose, CA, USA). Digitized traces were filtered with a 10-Hz low-pass Bessel function, leak corrected and analyzed for peak amplitude off-line using Clampfit v8 software (Molecular Devices).

## Zinc inhibition

GABA$_A$ receptors formed from α and β subunits alone are inhibited by low micromolar zinc, while αβγ receptors are insensitive to zinc [19]. We tested whether γ2L subunits were incorporated into oocyte-expressed wild-type GABA$_A$ receptors by assessing inhibition by 100 µM zinc. Voltage-clamped oocytes expressing receptors were first maximally activated with 3 mM GABA for 20 s, followed by a 5 min wash in ND96. Oocytes were then exposed to 3 mM GABA + 100 µM ZnCl$_2$ for 20 s, followed by another 5 min wash and then another 20 s exposure to 3 mM GABA. Currents were leak corrected, and the peak current with GABA + zinc was normalized to the average of the two GABA controls. Zinc inhibition was measured in 5 oocytes and rounded to the nearest percentage.

## GABA concentration-responses

Wild-type and mutant receptors were characterized for sensitivity to GABA. Voltage-clamped oocytes were exposed to ND96 solutions containing GABA at concentrations ranging from 0.1 µM to 30 mM for 30s, followed by 5 minute washes in ND-96 solution. Normalizing responses to maximal GABA were recorded every other trace. Responses to each GABA concentration were normalized to the average of the preceding and following maximum GABA responses. For each mutant, concentration-responses were assessed in five oocytes. Fractional normalized currents were fit with logistic equations (Eq. 1) using non-linear least squares:

$$\frac{I}{I_{max}} = \frac{1}{1 + 10^{(\log EC50 - \log[GABA]) * nH}}$$

(1)

EC$_{50}$ is the GABA concentration eliciting half maximal response and nH is the Hill coefficient (slope) of the relationship.

## GABA EC$_5$ enhancement by pTFD-di-iPr-BnOH

To determine whether pTFD-di-iPr-BnOH binds and modulates the cysteine mutant receptors, we assessed enhancement of currents elicited with EC$_5$ GABA (eliciting 5% of maximal GABA response) supplemented with 10 µM

pTFD-di-iPr-BnOH. Voltage-clamped oocytes expressing wild-type or mutant GABA$_A$ receptors were exposed for 30 s to EC$_5$ GABA in ND96, followed by a 30–60 s exposure to ND96 containing EC$_5$ GABA plus 10 μM pTFD-di-iPr-BnOH. After leak correction, enhancement was calculated as the ratio of peak current elicited with EC$_5$ GABA + 10 μM pTFD-di-iPr-BnOH to the peak current elicited by EC$_5$ GABA alone. Enhancement ratios were measured in 5 oocytes each for wild-type and the 8 mutant receptors.

## Effects of cysteine modification with pCMBS

Cysteine mutant receptors were characterized to establish how varying pCMBS exposure conditions affected GABA sensitivity, and to establish suitable conditions for substituted cysteine-modification-protection experiments. Voltage-clamped oocytes expressing mutant receptors were first activated with a maximally activating "High" GABA solution for 20 seconds, followed by a five minute wash in ND96. A second current sweep activated receptors with EC$_3$ "Low" GABA for 30 seconds, followed by another five minute ND-96 wash. This process was repeated, resulting in 4 sweeps, with each Low GABA peak current normalized to its preceding High GABA peak current. Oocytes exhibiting two consecutive Low/High GABA response ratios between 0.01 and 0.05 and differing by less than 20% were deemed to have stable responses suitable for modification experiments. Oocytes were then exposed to maximal GABA plus pCMBS at concentrations ranging from 1 μM to 100 μM, for durations ranging from 30 to 90 seconds, followed by a 20 minute ND96 wash. The pCMBS exposure was calculated as concentration×time (e.g. 100 μM × 30 s = 3000 μM × s). After pCMBS exposure and wash, oocytes were again twice activated with High GABA followed by Low (EC$_3$) GABA, with intervening ND-96 washes. The post-modification average Low/High GABA response ratio was calculated from these sweeps. Modification ratios were calculated as the ratio of post-modification Low/High GABA response ratio to the pre-modification Low/High GABA response ratio. The pCMBS exposures were varied to generate modification ratio vs. pCMBS exposure curves. Each oocyte was studied at a single pCMBS exposure condition. For each cysteine mutant, 5 oocytes were studied at 4 to 7 pCMBS exposure conditions. The modification ratio vs. pCMBS exposure curves were fit with mono-phasic or bi-phasic logistic equations similar to [Eq 1](), using non-linear least squares.

## Drug protection from pCMBS modification

Drug protection from pCMBS modification was assessed under pCMBS exposure conditions that produced approximately half of the maximal modification ratio for each mutant receptor, and a least a value of 3. This approach balanced the magnitude of functional modification signals (smaller with less pCMBS exposure) against the challenge of using a reversibly bound ligand to competitively block an irreversible covalent cysteine-modifying reaction. As described above, oocytes expressing mutant GABA$_A$ receptors were first exposed to pre-modification High and Low (EC$_3$) GABA solutions twice sequentially, to establish Low/High GABA response stability. Modification conditions for protection tests included pre-exposure to the test drug (3.2 μM etomidate, 8 μM R-mTFD-MPAB or 10 μM pTFD-di-iPr-BnOH) for 30 s followed by exposure to maximal GABA + test drug + pCMBS at the same concentration and duration conditions used for control modification. After a 20 minute wash, post-modification High and Low GABA responses were again assessed in duplilcate and a modification ratio was calculated. Each oocyte was used in a single SCAMP experiment. Protection by pTFD-di-iPr-BnOH was tested in all 8 cysteine-substituted receptors. Etomidate was used as a positive control for anesthetic protection in α1M236C and β3M286C mutants and as a negative control in α1S270C, α1A291C, β3L231C, γ2S301C, γ2I242C, and γ2L246C mutants. R-mTFD-MPAB was used as a positive control in α1S270C, α1A291C, β3L231C, and γ2LS301C mutants and as a negative control in α1M236C and β3M286C mutants. For each cysteine mutant and protective drug combination, five oocytes were studied.

## Statistical analysis

Non-linear least squares fits and statistical analyses were performed in Prism 10.4.0. (GraphPad Software, San Diego, CA, USA). Fitted GABA EC$_{50}$s are reported as mean with 95% confidence interval. Other results are reported and displayed as mean ± SD. Single variable ANOVA analysis and Dunnett's post-hoc tests were used to compare

pTFD-di-iPr-BnOH enhancement of $EC_5$ GABA responses in each mutant to that in wild-type receptors. For each mutant receptor, single variable ANOVA analysis was also used to compare modification ratios measured after modification in the presence of anesthetics against control (GABA + pCMBS) conditions. Protection (steric inhibition of pCMBS modification) was inferred when anesthetics produced significant reductions in modification ratios relative to control (no anesthetic) values. The statistical significance threshold was $p < 0.05$.

## Results

### Zinc inhibition of α1β3γ2L GABA$_A$ receptors

Five oocytes expressing wild-type α1β3γ2L GABA$_A$ receptors were studied to assess inhibition by 100 μM zinc. The percentages of inhibition were, in descending order of magnitude, 9, 8, 5, 5, and 4 (mean ± SD = 6.2 ± 2.2%).

### Functional characterization of cysteine-substituted mutant α1β3γ2L GABA$_A$ receptors

*Xenopus* oocytes heterologously expressing subunit mRNA mixtures encoding wild-type and 8 single-residue cysteine mutant GABA$_A$ receptors of interest were functionally characterized using two-microelectrode voltage clamp electrophysiology. Under voltage clamp, all eight mutant receptors produced inward currents in response to GABA in a concentration-dependent manner (S1 Fig). Logistic analysis of normalized currents resulted in a range of GABA $EC_{50}$s for different mutants, from 6.2 to 260 μM (Table 1). Additionally, all eight mutants exhibited positive modulation of $EC_5$ GABA currents in the presence of 10μM pTFD-di-iPr-BnOH. Six of the mutants displayed positive modulation by pTFD-di-iPr-BnOH similar to that observed in wild-type α1β3γ2L receptors (range, 5 to 10-fold) while two mutations, β3L231C and β3M286C, conferred significantly more pTFD-di-iPr-BnOH modulation than wild-type (both $p < 0.001$ based on ANOVA analysis with Dunnett's post-hoc tests; Table 1).

### Functional effects of pCMBS exposure on cysteine-substituted mutant GABA$_A$ receptors

Control electrophysiological experiments exposing wild-type GABA$_A$ receptors to up to 100 μM pCMBS for 90s produced no significant change in Low/High GABA response ratios (Table 1, n = 5). In contrast, all 8 cysteine-substituted mutant receptors displayed persistently larger Low/High GABA response ratios after exposure to GABA + pCMBS. Fig 2A shows example currents recorded from an oocyte expressing α1A291Cβ3γ2L receptors. In 7 mutant receptors, Low/High GABA modification ratios rose monotonically as pCMBS exposure increased, whereas increasing pCMBS exposure produced a biphasic effect in receptors with α1S270C mutations, with a maximum effect at 300 μM x s (S2 Fig). Maximum pCMBS modification ratios varied among the mutants, ranging from 5-fold to 17-fold (Table 1).

### Anesthetic protection from cysteine modification

Control modification conditions for anesthetic protection studies in each of the mutant receptors (Table 1) were chosen to produce the half-maximal modification ratio or at least a value of 3. For α1S270Cβ3γ2L receptors that showed biphasic modification effects, control modification conditions were exposure to GABA + 90 μM x s pCMBS (1 μM x 90s), producing about half-maximal effects in the lower exposure range.

Fig 2 shows current sweeps recorded from four oocytes expressing α1A291Cβ3γ2L under control modification conditions (Fig 2A) and in the presence of etomidate, R-mTFD-MPAB, or pTFD-di-iPr-BnOH (Figs 2B, 2C and 2D, respectively). In each case, the pre-modification Low/High GABA response ratio was near 0.03. The post-modification Low/High GABA response ratio in the control example (Fig 2A) was 0.12, resulting in a modification ratio of 4.0 for this oocyte. In the presence of etomidate (Fig 2B), the modification ratio was 3.8, close to control. In oocytes expressing α1A291Cβ3γ2L receptors co-exposed to R-mTFD-MPAB (Fig 2C) or pTFD-di-iPr-BnOH (Fig 2D) during pCMBS modification, the modification ratios were, respectively, 2.1 (47% lower than control) and 2.7 (33% lower than control).

**Table 1. Cysteine Mutant GABA$_A$ Receptor Properties.**

| Cysteine Mutant (Interfacial Locus) | GABA EC$_{50}$ (µM) [95% CI] | pTFD-di-iPr-BnOH Enhancement Ratio Mean + SD | Maximal pCMBS Modification Ratio Mean ± SD | Modification Conditions for Protection Studies (pCMBS µM x s) |
|---|---|---|---|---|
| α1β3γ2L | 41 [28-58] | 7 ± 2.3 | 1.1 ± 0.21 | NA |
| **α1M236C**β3γ2L (α–/M1) | 240 [210-270] | 9 ± 2.4 | 13 ± 1.4 | 10 µM x 30 s |
| **α1S270C**β3γ2L (α+/M2) | 36 [32 –40] | 7 ± 1.7 | 11.8 ± 0.77 | 1 µM x 90 s |
| **α1A291C**β3γ2L (α+/M3) | 45 [40-50] | 7 ± 3.5 | 4.7 ± 0.61 | 10 µM x 90 s |
| α1**β3L231C**γ2L (β–/M1) | 35 [28 –43] | 23 ± 2.2* | 13 ± 1.3 | 100 µM x 30 s |
| α1**β3M286C**γ2L (β+/M3) | 260 [220-320] | 43 ± 9.1* | 9 ± 3.1 | 1 µM x 30 s |
| α1β3**γ2I242C** (γ–/M1) | 23 [20–26] | 5 ± 1.0 | 17 ± 4.9 | 100 µM x 30 s |
| α1β3**γ2L246C** (γ–/M1) | 6.2 [5.4-7.1] | 7 ± 2.0 | 5 ± 1.0 | 100 µM x 30 s |
| α1β3**γ2S301C** (γ+/M3) | 39 [33 –44] | 10 ± 2.1 | 5.3 ± 0.23 | 100 µM x 90 s |

Interfacial loci indicate the subunit face and transmembrane helix where the mutated residue is located on structural models (Fig 1). pTFD-di-iPr-BnOH enhancement is the ratio of peak currents stimulated with EC$_5$ GABA plus 10 µM pTFD-di-iPr-BnOH vs. EC$_5$ GABA alone. Modification ratios are the ratio of post-modification to pre-modification Low/High GABA response ratios. *Significantly different from wild-type (p < 0.001) by ANOVA with post-hoc Dunnett's tests. pCMBS = para-chloromercuribenzenesulfonate..

Figs 3–5 display control modification and anesthetic protection results (modification ratio mean ± SD; n = 5 each) for the 8 cysteine-substituted mutant receptors that were studied, respectively for α1, β3, and γ2L subunits. For each mutant receptor, modification ratios observed when anesthetics were present during modification were compared to control modification ratios. Protection from pCMBS modification was inferred when an anesthetic significantly reduced modification ratios relative to control. In oocytes expressing α1M236Cβ3γ2L receptors (Fig 3A), protection was inferred with etomidate, but not with R-mTFD-MPAB or pTFD-di-iPr-BnOH. In oocytes expressing α1S270Cβ3γ2L receptors (Fig 3B), protection was inferred with R-mTFD-MPAB, but not with etomidate or pTFD-di-iPr-BnOH. In oocytes expressing α1A291Cβ3γ2L receptors (Fig 3C), protection was inferred with R-mTFD-MPAB and pTFD-di-iPr-BnOH, but not with etomidate. In oocytes expressing α1β3L231Cγ2L receptors (Fig 4A), protection was also inferred with R-mTFD-MPAB and pTFD-di-iPr-BnOH, but not with etomidate. In oocytes expressing α1β3M286Cγ2L receptors (Fig 4B), protection was inferred with etomidate, but not with R-mTFD-MPAB or pTFD-di-iPr-BnOH. In oocytes expressing α1β3γ2I242C receptors (Fig 5A), no protection was inferred with etomidate or pTFD-di-iPr-BnOH. In oocytes expressing α1β3γ2L246C receptors (Fig 5B), protection was inferred with pTFD-di-iPr-BnOH, but not with etomidate. In oocytes expressing α1β3γ2S301C receptors (Fig 5C), protection was inferred with R-mTFD-MPAB, but not with etomidate or pTFD-di-iPr-BnOH.

## Discussion

We tested the steric interactions of pTFD-di-iPr-BnOH with all five outer transmembrane interfacial pockets of α1β3γ2L GABA$_A$ receptors using SCAMP at 8 residues. Our results (Figs 3 through 5) showed evidence of pTFD-di-iPr-BnOH protection at α1A291C (α1+), β3L231C (β3–), and γ2L246C (γ2–). We found no evidence of pTFD-di-iPr-BnOH protection at α1M236C (α1–), β3M286C (β3+) or γ2S301 (γ2+). These results support the hypothesis that pTFD-di-iPr-BnOH binds

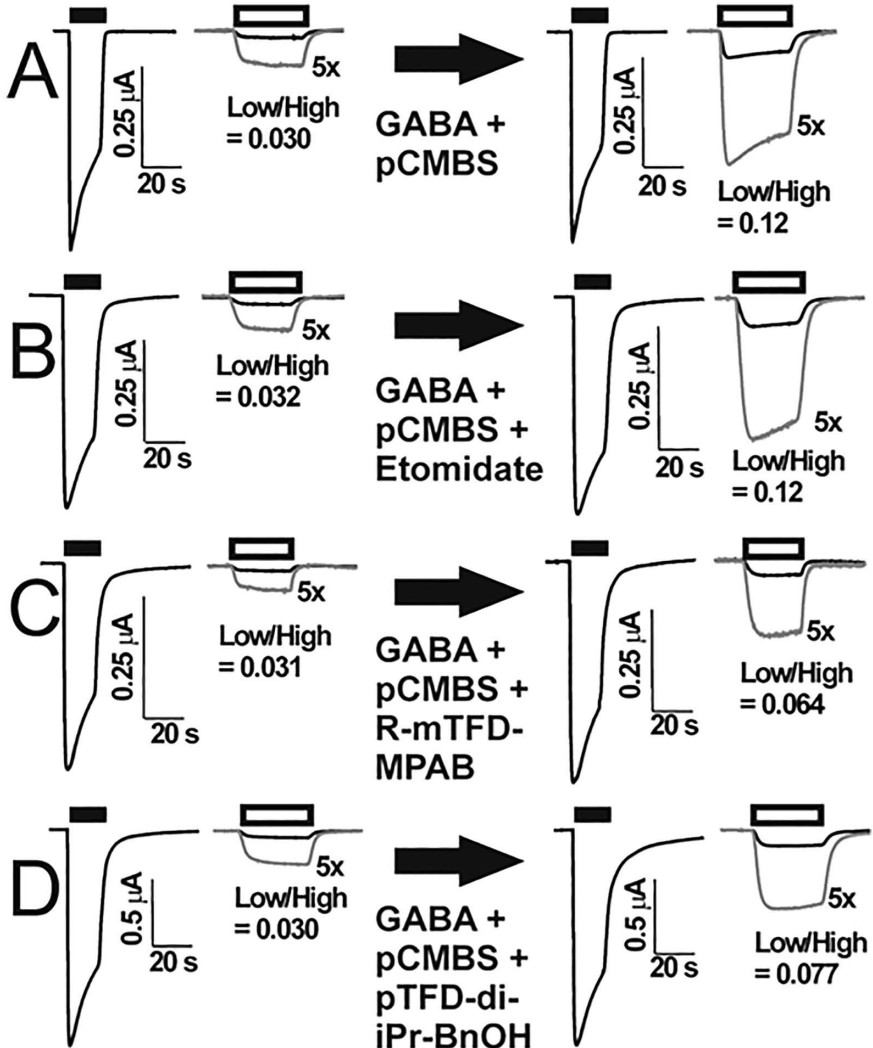

**Fig 2. Substituted Cysteine Modification and Anesthetic Protection (SCAMP) in α1A291Cβ3γ2L GABA_A Receptors.** Each row depicts four voltage-clamped current traces recorded from a single oocyte expressing α1A291Cβ3γ2L GABA_A receptors both before and after exposure to pCMBS alone (A; control) or together with one of the tested anesthetics (B-D; protection tests). Modification conditions are indicated under the large arrows between pairs of traces. Bars above traces represent applications of high GABA (3 mM; black bars) and low GABA (EC_3 = 4 µM; white bars). The two traces on the left show pre-modification current responses and the right-side pair of traces show post-modification responses. Grey traces are low GABA currents magnified five times to aid visual comparisons. The Low/High GABA response ratio for each pair of traces is shown beneath the low GABA traces. **Row A** shows results of a control modification exposure to 3 mM GABA + 10 µM pCMBS for 90 s. The modification ratio is the ratio of the post-modification to pre-modification Low/High GABA response ratios: 0.12/0.03 = 4.0. **Row B** shows results for another oocyte under control modification conditions with addition of 3.2 µM etomidate. The modification ratio is 0.12/0.032 = 3.8. **Row C** shows results for another oocyte under control modification conditions with addition of 8 µM R-mTFD-MPAB. The modification ratio is 0.064/0.031 = 2.1. **Row D** shows results for another oocyte under control modification conditions with addition of 10 µM pTFD-di-iPr-BnOH. The modification ratio is 0.077/0.030 = 2.7. pCMBS = para-chloromercuribenzenesulfonate.

selectively in α+/β– and α+/γ– outer transmembrane interfacial pockets, based on previous subunit-level photolabeling and mutant-function results [15].

To confidently infer that positive SCAMP results indicated steric interactions between protective ligands and cysteine-substituted residues, we established protection-sensitive control modification conditions for each mutant receptor and

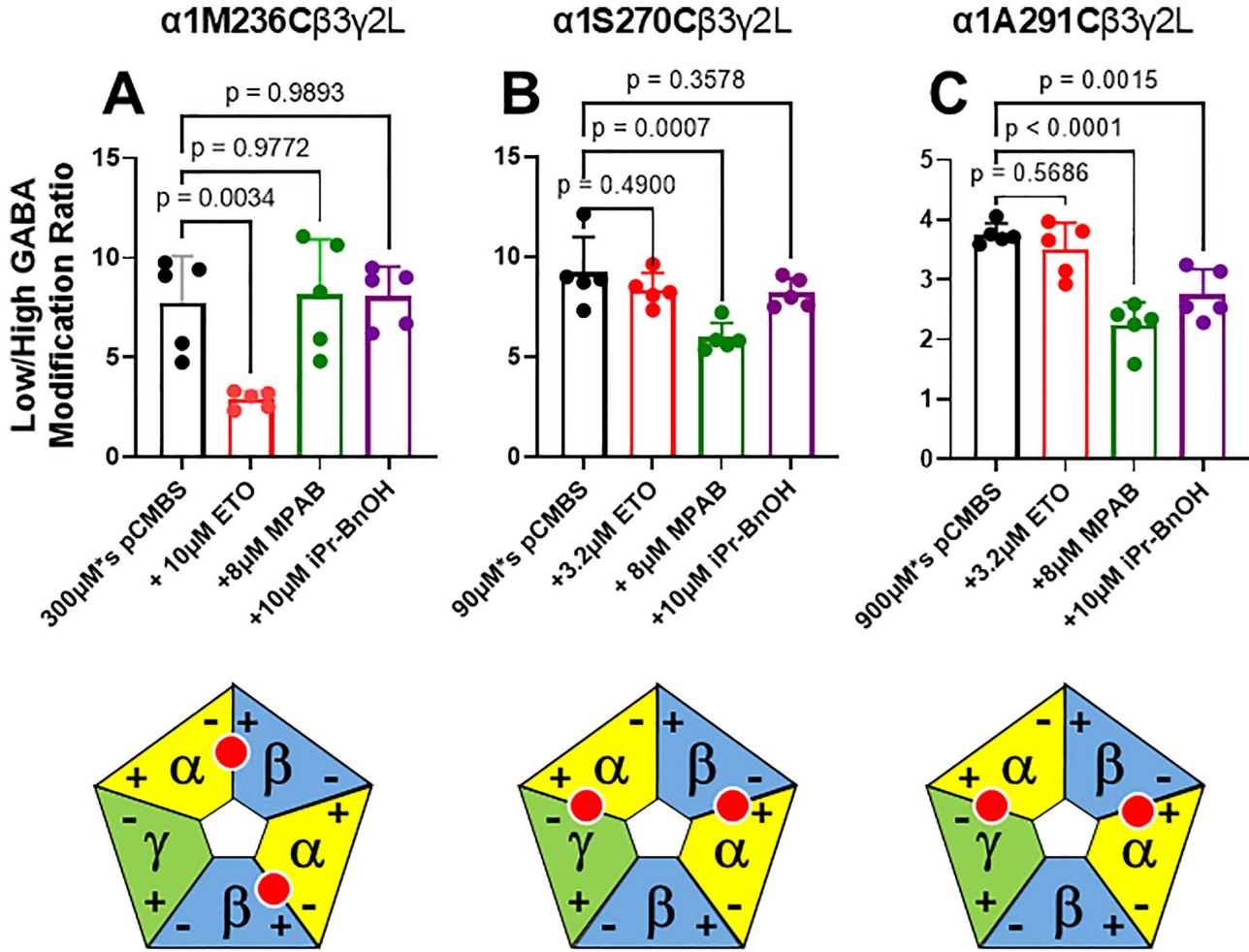

**Fig 3. Summary of Substituted Cysteine Modification and Anesthetic Protection Studies in α1 Subunits.** Each panel shows a bar graph (mean±SD) and individual oocyte results (solid circles) summarizing the modification ratio results for oocytes expressing a cysteine-substituted mutant GABA$_A$ receptor under different modification conditions, labeled along the x-axis (n=5 per condition). For each mutant receptor, control modification ratio results (black) were compared to those from receptors modified in the presence of anesthetics (etomidate=red; m-TFD-MPAB=green; pTFD-di-iPr-BnOH=purple) using single-variable ANOVA analysis with Dunnett's post-hoc tests. Analysis results are shown as p-values for the relevant pairs. Protection was inferred when an anesthetic significantly reduced modification ratios (p<0.05) relative to control. Below each bar graph is a diagram showing the subunit interfaces where the mutation is located. **Panel A:** α1M236C (α–) was protected by etomidate. **Panel B:** α1S270C (α+) was protected by R-mTFD-MPAB. **Panel C:** α1A291C (α+) was protected by both R-mTFD-MPAB and pTFD-di-iPr-BnOH. ETO=etomidate; MPAB=R-mTFD-MPAB; iPr-BnOH=pTFD-di-iPr-BnOH; pCMBS=para-chloromercuribenzenesulfonate.

performed essential control experiments. Sensitivity to protection in SCAMP requires evidence both that the protective ligand (pTFD-di-iPr-BnOH) bound to mutant receptors and that pCMBS modification produced robust irreversible functional effects. Indeed, all 8 mutant receptors displayed reversible positive modulation by pTFD-di-iPr-BnOH comparable to or greater than that in wild-type receptors (Table 1), indicating action *via* the same binding sites in wild-type and mutant receptors. Exposing wild-type receptors to pCMBS+GABA produced no irreversible functional effects. Thus, modification of native cysteines either did not occur or did not alter GABA sensitivity, consistent with previous findings [12,20,21]. Exposure of all mutant receptors to pCMBS resulted in persistently increased GABA sensitivity, indicating covalent

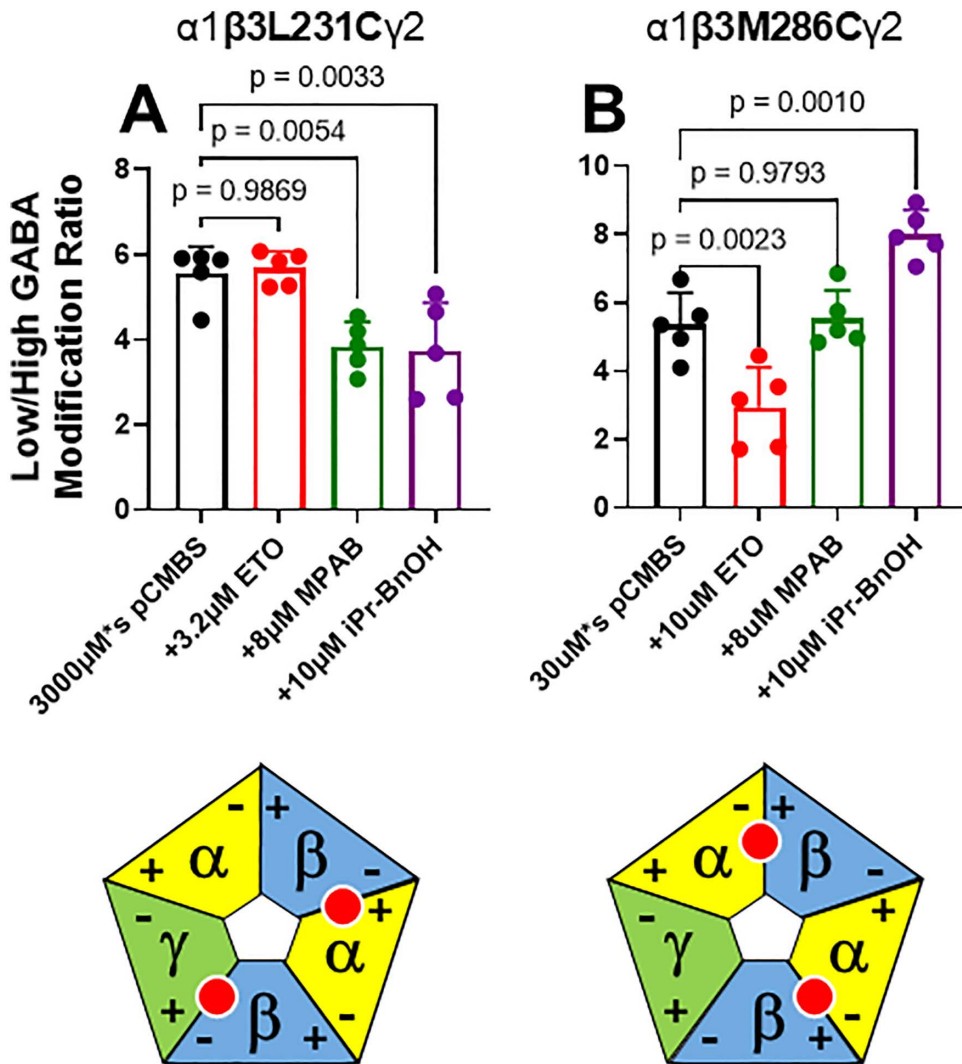

**Fig 4. Summary of Substituted Cysteine Modification and Anesthetic Protection Studies in β3 Subunits.** See Fig 3 legend for detailed description. Protection was inferred when an anesthetic significantly reduced modification ratios (p < 0.05) relative to control. Below each bar graph is a diagram showing the interfaces where the mutation is located. **Panel A:** β3L231C (β–) was protected by both R-mTFD-MPAB and pTFD-di-iPr-BnOH. **Panel B:** β3M286C (β+) was protected by etomidate. ETO = etomidate; MPAB = R-mTFD-MPAB; iPr-BnOH = pTFD-di-iPr-BnOH; pCMBS = para-chloromercuribenzenesulfonate.

modification. Because SCAMP depends on competition between a non-covalent ligand and the covalent modifier pCMBS, we established control conditions that approximated the rate of sulfhydryl modification in each mutant receptor. These conditions varied widely, possibly reflecting differential pCMBS and/or aqueous access to the mutated residues. We also included positive and negative control SCAMP studies utilizing two anesthetic modulators with well-established binding selectivity for either outer transmembrane β+/α– interfacial pockets (etomidate) or homologous α+/β– and γ+/β– pockets (R-mTFD-MPAB). We excluded the β3N265C (β+) mutation from our study because etomidate binds extremely weakly to α1β2N265Cγ2L receptors and fails to protect at the tested concentration, undermining SCAMP interpretation [22].

Another precondition for interpretation of pTFD-di-iPr-BnOH protection results is establishing that receptors we studied contain α1, β3, and γ2L subunits in the stoichiometry and arrangement shown in Fig 1. We showed that our

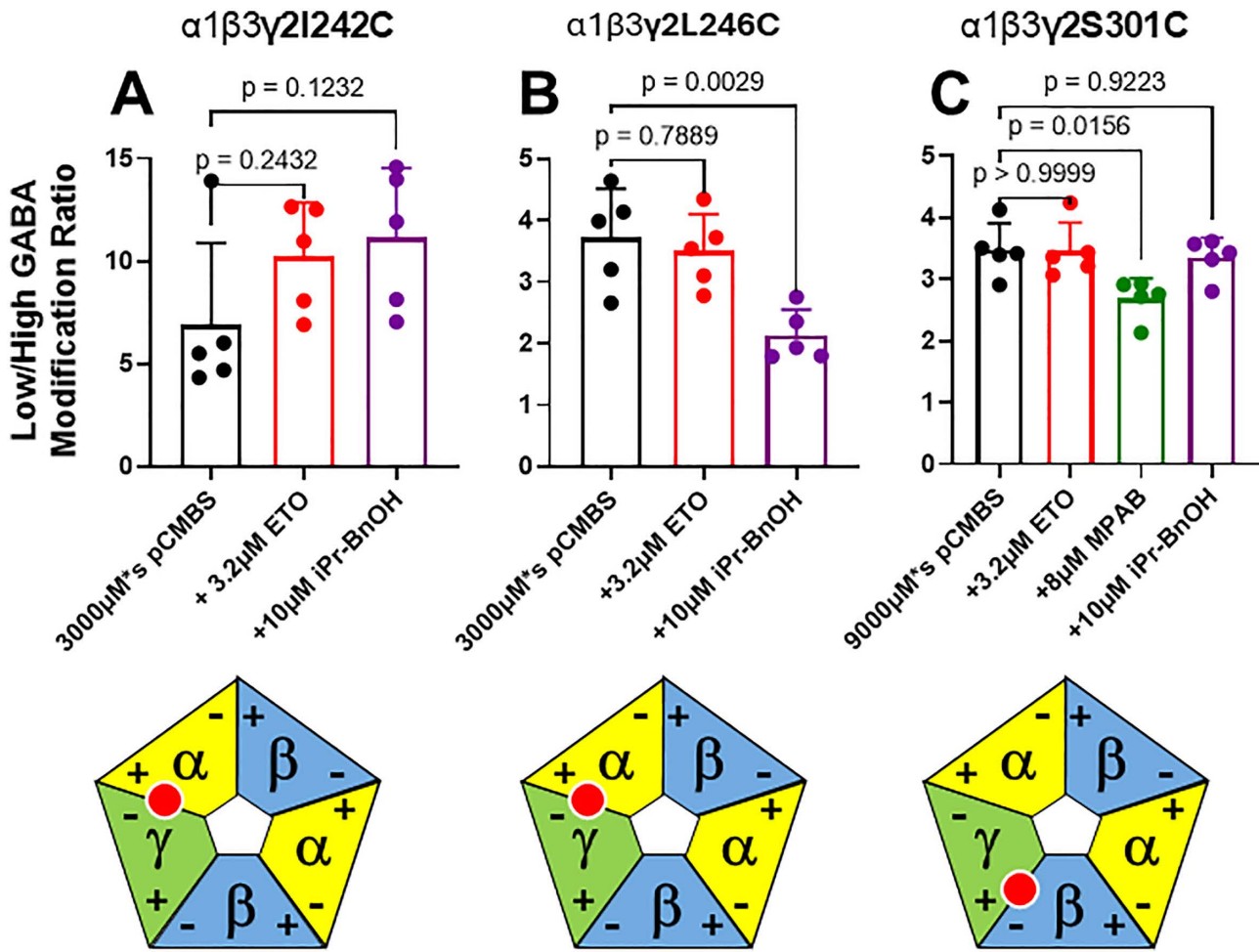

**Fig 5. Summary of Substituted Cysteine Modification and Anesthetic Protection Studies in γ2L Subunits.** See Fig 3 legend for detailed description. Etomidate was used as the sole negative control drug for the two γ– mutants (panels A and B) because its binding sites are best established. Protection was inferred when an anesthetic significantly reduced modification ratios (p < 0.05) relative to control. Below each bar graph is a diagram showing the interface where the mutation is located. **Panel A:** γ2I242C (γ–) was not protected. **Panel B:** γ2L246C (γ–) was protected by pTFD-di-iPr-BnOH. **Panel C:** γ2S301C (γ+) was protected by R-mTFD-MPAB. ETO = etomidate; MPAB = R-mTFD-MPAB; iPr-BnOH = pTFD-di-iPr-BnOH; pCMBS = para-chloromercuribenzenesulfonate.

oocyte-expressed wild-type receptors are insensitive to 100 μM zinc, indicating that over 90% of GABA-activated currents we recorded are from zinc-insensitive receptors containing γ2L subunits. In all experiments using mutated subunits, pCMBS exposure induced persistent functional changes, while none were seen in wild-type receptors. Thus, the irreversible effects of pCMBS exposure alone demonstrate incorporation of mutated subunits. Moreover, control protection studies with etomidate and R-mTFD-MPAB produced results fully concurring with previous results using SCAMP and other methods, including photolabeling and cryo-EM. Considered together, these control results indicate that in the mutated receptors, all 3 subunits were consistently incorporated and arranged as shown in Fig 1.

Table 2 summarizes our current SCAMP results along with previously reported SCAMP and mutant-function results for etomidate, R-mTFD-MPAB and pTFD-di-iPr-BnOH at 12 loci that abut the 5 homologous outer transmembrane inter-subunit pockets (Fig 1). As noted above, we found total agreement of our current SCAMP results with prior SCAMP results for etomidate and R-mTFD-MPAB [9,12,16], demonstrating robustness of this experimental approach. Etomidate

Table 2. Summary of Mutant-Function and SCAMP Results.

| | | α1− (M1) | | α1+ (M2, M3) | | β3− (M1) | | β3+ (M2, M3) | | γ2− (M1) | | γ2+ (M2, M3) | |
| --- | --- | --- | --- | --- | --- | --- | --- | --- | --- | --- | --- | --- | --- |
| | | L232 | M236 | S270 | A291 | M227 | L231 | N265 | M286 | I242 | L246 | S280 | S301 |
| Etomidate | Mutants | +[9, 12] | +[9, 12] | −[13, 15] | −[13] | −[9, 15] | +[9] | +[13, 15] | +[13] | −[9, 15] | −[9] | −[13, 15] | −[13] |
| | SCAMP | +[9] | +[9], * | −* | −* | −[9] | −[9], * | ** | +[16], * | −[9],6 | −[9], * | ND | −* |
| R-mTFD-MPAB | Mutants | −[9] | −[9] | +[13, 15] | +[13] | +[9, 15], | −[9] | −[13, 15] | −[13] | −[9, 15] | −[9] | +[13, 15] | +[13] |
| | SCAMP | −[9] | −[9], * | +* | +* | +[9] | +[9], * | ND | −* | −[9] | −[9] | ND | +* |
| pTFD-di-iPr-BnOH | Mutants | ND | ND | +[15] | ND | +[15] | ND | −[15] | ND | +[15] | ND | −[15] | ND |
| | SCAMP | ND | −* | −* | +* | ND | +* | ND | −* | −* | +* | ND | −* |

Positive results, suggesting steric proximity for anesthetic-residue pairs, are indicated by "+" symbols and shaded green. Negative results are indicated by "−" symbols. ND indicates no data. Mutant-function studies are deemed positive when bulky mutations (mostly tryptophans) at the site significantly reduce positive modulation by anesthetics. SCAMP is Substituted Cysteine Modification and Protection and is positive when adding anesthetic to cysteine modifier significantly reduces persistent post-exposure effects due to covalent modification. Bracketed numbers in the table are reference citations. *=Results of this study. **β3N265C is insensitive to etomidate, making SCAMP uninformative [22].

selectively protected modifiable cysteines in α1− and β+ interfaces, but not in α1+, β3−, γ2−, or γ2+. R-mTFD-MPAB protected modifiable cysteines in α1+, β3−, and γ2+, but not α1−, β+, or γ2−. These results also agree with azi-etomidate and R-mTFD-MPAB photolabeling [11,14] and cryo-EM studies of etomidate and phenobarbital [3]. Our SCAMP results with pTFD-di-iPr-BnOH show selective protection in α+, β−, and γ2− interfaces, echoing prior mutant-function results [15]. Table 2 also reveals some non-concordance for mutant-function vs. SCAMP results with pTFD-di-iPr-BnOH. The α1S270I and γ2I242W mutations both significantly reduced receptor modulation by pTFD-di-iPr-BnOH [15], but protection from pCMBS modification at α1S270C and γ2I242C sulfhydryls was not observed. Non-concordant results were also previously found when comparing mutant-function and SCAMP results for etomidate and R-mTFD-MPAB (both at β3L231). Importantly, SCAMP results reported by Nourmahnad et al [9] were fully concordant with photolabeling results for azi-etomidate and R-mTFD-MPAB, while mutant-function results were not. Thus, in the absence of photolabeling or cryo-EM data for pTFD-di-iPr-BnOH, we weigh the SCAMP results as more accurate than mutant-function results. Considering together both mutant-function and SCAMP results for pTFD-di-iPr-BnOH, both approaches support selective binding in α+/β− and α+/γ− transmembrane interfacial pockets that are homologs of the etomidate and R-mTFD-MPAB sites.

Our results also suggest that while pTFD-di-iPr-BnOH and R-mTFD-MPAB both bind in the α+/β− transmembrane interface, their binding sites may not fully overlap. This is based on the differential SCAMP results observed with the α1S270C mutation. Because SCAMP protection depends on the size of the cysteine modifying reagent [21], we hypothesize that R-mTFD-MPAB binds closer to α1S270 than does pTFD-di-iPr-BnOH.

While additional mutant-function and/or SCAMP studies might suggest more details about the pTFD-di-iPr-BnOH binding sites, neither of these approaches are unbiased. Amino-acid level photolabeling analysis and/or cryo-EM structural studies of pTFD-di-iPr-BnOH in α1β3γ2L GABA$_A$ receptors are needed to more confidently establish its GABA$_A$ receptor binding sites.

The results summarized in Table 2 indicate that all five homologous outer transmembrane interfacial pockets formed by α1β3γ2L GABA$_A$ receptors are positive allosteric modulator/co-agonist binding sites: etomidate selectively binds in two β+ interfaces, R-mTFD-MPAB selectively binds in two β− interfaces, and pTFD-di-iPr-BnOH selectively binds in two α+ interfaces, as illustrated in Fig 1. Of particular interest is evidence supporting pTFD-di-iPr-BnOH binding within the α+/γ− outer transmembrane interface, which was previously labeled an "orphan" anesthetic site [9]. A photoreactive analog of isoflurane, azi-ISO, was reported to incorporate at γ2Y241, immediately adjacent to γ2I242, and at α1P278 and α1S276, both located about two helical turns extracellular to α1S270. [23] These photo-incorporation loci suggest azi-ISO binding at sites distinct from the pTFD-di-iPr-BnOH α+/γ− interfacial pocket we probed or possibly in different parts of a large contiguous pocket.

While the anesthetic compounds we studied all bind selectively in two transmembrane interfaces per receptor, it is conceivable that sedative-hypnotic drugs may be developed that selectively bind to only one of the three unique interfacial anesthetic sites formed by α1β3γ2L GABA$_A$ receptors: α+/γ–, α+/β–, or γ+/β–. Where heterogeneous residues are found among subunit isotypes, such compounds might also display unique pharmacological actions based on selective actions on different GABA$_A$ receptor subtypes [24,25].

Our study had several limitations. We did not attempt to differentiate the two β+/α– interfacial pockets, which could have been achieved using concatenated subunit constructs. However, prior mutant function studies with etomidate suggest they are functionally equivalent [26]. We did not study all 12 loci in Table 2. Instead, we limited our study to 8 mutants that tested pTFD-di-iPr-BnOH interactions at all 5 interfacial pockets, with two each in the interfacial site of greatest interest: α+/γ–. This approach complemented and was also more thorough than the previous mutant-function studies, which included only 5 loci [15]. Other mutant studies have shown allosteric cross-talk between β+ and β– anesthetic binding pockets in α1β3γ2L GABA$_A$ receptors [13]. However, our control studies and prior SCAMP studies [9] have found no evidence of allosteric interactions producing false-positive results, although SCAMP may result in false negatives when the size of modifying probes is too small or too large [21] or the cysteine substitution dramatically reduces drug binding affinity [22].

In conclusion, our current SCAMP results significantly strengthen evidence for the anesthetic photolabel pTFD-di-iPr-BnOH acting selectively *via* synaptic GABA$_A$ receptor α+/β– and α+/γ– outer transmembrane interfacial pockets that are homologs of etomidate β+/α– binding sites. Amino-acid level photolabeling and/or cryo-EM structural evidence is needed to further test this conclusion.

## Supporting information

**S1 Fig. GABA Concentration Responses of Cysteine Substituted GABA$_A$ Receptors.**
(PDF)

**S2 Fig. Functional Effects of Varying pCMBS Exposures in Cysteine Substituted Mutant α1β3γ2L GABA$_A$ Receptors.**
(PDF)

## Acknowledgments

We thank Prof. Keith W. Miller (Massachusetts General Hospital Dept. of Anesthesia Critical Care & Pain Medicine, Boston, MA, USA) and Prof. Karol Bruzik (Department of Medicinal Chemistry and Pharmacognosy, University of Illinois, Chicago, USA) for providing us with R-mTFD-MPAB and pTFD-di-iPr-BnOH. We thank Dr. Chen Chen (Dept. of Anesthesiology, First People's Hospital of Changzhou, China, research fellow at Massachusetts General Hospital Dept. of Anesthesia Critical Care & Pain Medicine, Boston, MA, USA) for assistance with zinc experiments.

## Author contributions

**Conceptualization:** Stuart A. Forman.

**Data curation:** Kieran Bhave.

**Formal analysis:** Kieran Bhave, Stuart A. Forman.

**Funding acquisition:** Stuart A. Forman.

**Investigation:** Kieran Bhave.

**Methodology:** Kieran Bhave, Stuart A. Forman.

**Project administration:** Stuart A. Forman.

**Visualization:** Stuart A. Forman.

**Writing – original draft:** Kieran Bhave.

**Writing – review & editing:** Kieran Bhave, Stuart A. Forman.

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
