## [Decision Letter · Decision Letter 0]

12 Aug 2025

Dear Dr. Forman,

Thank you for submitting your manuscript to PLOS ONE. After careful consideration, we feel that it has merit but does not fully meet PLOS ONE’s publication criteria as it currently stands. Therefore, we invite you to submit a revised version of the manuscript that addresses the points raised during the review process.  Both reviewers believe that the study provides important information, but raise a number of points that the authors should address in a revised version.  Addressing these point is expected to improve the quality of the manuscript.

We look forward to receiving your revised manuscript.

Kind regards,

Uwe Rudolph

Academic Editor

PLOS ONE

Journal Requirements:

“SAF received supported for this work from a grant from the US National Institutes of Health (R35GM141951) and the Dept. of Anesthesia Critical Care & Pain Medicine, Massachusetts General Hospital, Boston, MA, USA.”

Reviewers' comments:

Reviewer's Responses to Questions

**Comments to the Author**

1. Is the manuscript technically sound, and do the data support the conclusions?

Reviewer #1: Yes

Reviewer #2: Partly

2. Has the statistical analysis been performed appropriately and rigorously?

Reviewer #1: Yes

Reviewer #2: Yes

3. Have the authors made all data underlying the findings in their manuscript fully available?

Reviewer #1: Yes

Reviewer #2: No

4. Is the manuscript presented in an intelligible fashion and written in standard English?

Reviewer #1: Yes

Reviewer #2: Yes

Reviewer #1: The manuscript "Substituted cysteine modification and protection indicates selective interactions of the anesthetic photolabel pTFD-di-iPr-BnOH with α+/β- and α+/γ- transmembrane subunit interfaces of synaptic GABAA receptors" by Bhave and Forman investigates the interfaces between adjacent subunits of GABA-A receptors at which the anesthetic photolabel pTFD-d-iPr-BnOH binds, and compares those results to those found with two other agents, etomidate and R-mTFD-MPAB. All of these compounds, which act as positive allosteric modulators of GABA-A receptors, are shown to possess unique inter-subunit binding sites. The studies appear to have been painstakingly carried out and the manuscript is written in a clear and straightforward manner, although I do have a suggestion that might improve clarity.

I have no major concerns about this study and the conclusions reached, and my comments below are primarily suggestions that might improve it.

What happens to the degree of modulation produced by these agents if one omits the gamma2 subunit, only expressing alpha/beta heteromeric receptors? Presumably etomidate modulation would not change much but what happens to MPAB and BnOH modulation if the gamma subunit contribution to binding is removed? Would one be able to obtain information about the relative importance of the two intersubunit binding sites of MPAB and BnOH by comparing their responses on alpha1beta3gamma2 vs alpha1beta3 receptors? Data provided in the paper already suggest that this might be so. In Figure 3, cysteine mutants at the alpha (+) / beta (-) interface (Fig 3B-D) show marked effects of MPAB in blocking pCMBS binding, but the effect is considerably smaller at the gamma (+) / beta (-) interface (Fig 3H). In contrast, it looks like the gamma2 subunit plays a greater role in the effects of BnOH, since mutations at alpha1 A291C and gamma2 L246C seem to be similar in terms of BnOH protection against pCMBS binding.

Why was MPAB not tested in Figs 3F (gamma2 I242C) and 3G (alpha1 L246C), if only for the sake of completeness?

The location of the gamma2 I242 residue is not depicted in Fig. 1 and no data in Fig 3 is provided for the gamma2 S280 mutation to cysteine that is shown in the gamma subunit in Fig 1. This should be made consistent, or at least explained.

In the Discussion it might be worth highlighting that, even though both MPAB and BnOH act at the alpha (+) / beta (-) interface, they do so differently. In Fig 3 it is shown that there is a difference between MPAB and BnOH in their effects on pCBMS enhancement in the alpha1S270C mutant (only MPAB blocked), compared to the alpha1A291C and beta3L231C mutants (both blocked), so the binding of these agents at this interface may overlap but is not identical at this site. This is depicted in Table 2, but may still be worth highlighting in the Discussion.

Something for the authors to consider in a follow-up study: Does BnOH binding to the alpha(+)/gamma(-) site interfere with benzodiazepine actions of these receptors?

The authors should consider breaking up Figure 3 into three figures, with each new figure focused on alpha, beta or gamma subunit mutants. Each figure should also contain a small version of the graphic currently shown in Fig. 1, pointing out the mutations and subunit interfaces being dealt with in that figure. In reviewing this manuscript, I was constantly going back and forth between Figs. 1 and 3, trying to keep things straight. Maybe the current Fig 1. could then become the last figure in the paper, that summarizes the findings.

Minor typo

line 283, "findinds"

Reviewer #2: Bhave and Forman present a study concerned with the binding sites of a photoreactive compound “pTFD-di-iPr-BnOH“, in this review called pTFD for short. They use a cysteine based approach called SCAMP, and conclude that pTFD interacts with two sites at a1b2g2 receptors – namely the a1+/b2- and a1+/g2- binding sites in the TMD that are homologous with the “etomidate- site“ at b2+/a1-. The experimental approach has been used in the past and is thus well established. The results are of interest. The reasoning behind the study design, details in the methods, and the presentation and interpretation of the results can and should be improved for more clarity as indicated point by point below. Most suggested changes concern data analysis and presentation of the structural aspects and the results, but some additional control experiments are also needed.

Introduction:

Figure 1 is meant to support the Introduction and to provide information about the localization of the amino acids that were mutated to Cys in this and previous work. It depicts a schematic pentamer with the position of amino acids involved in the binding sites of interest. This schematic view is somewhat misleading, as it suggests that all three amino acids that line a given binding site are in the same plane oft he TMD – which is actually not the case. For unknown reasons, the authors selected a M1 position which is further towards the intracellular end compared to those in M1 and M3. A rendering of an available experimental structure to show the positions of the mutated amino acids would be more informative compared to the presented scheme. If the authors prefer the schematic rendering, they should clearly state that the amino acids are in different planes relative to the ligand position. Moreover, the reader would be helped by a rendering with all 12 amino acids that are displayed in table 2.

The explanation in lines 51/52 on the “+“ and “-“ sides of the binding sites is lacking M2, which contributes to both. A more complete leged for Figure 1 would solve this problem.

Legend for Fig 1: What is a “subsite“, what are subsite selective compounds???

Methods and study design:

The reasoning behind the selection of the amino acids is not provided at all. See also the remark concerning Figure 1, the choice of the amino acid on M1 (- side) is non- obvious. While the reader learns in the discussion why table 2 features 12 amino acids, figure 1 displays nine, and the results from 8 mutants (partly displayed in Fig 1) are discussed, the reader would be helped if a table provides clarity, or if table 1 is extended to reflect the whole situation that is also summarized in table 2. For gamma2 minus, two mutants were studied, which is reflected correctly in the methods and in the results chapter. While g2L246 (displayed in Fig 1) ist he homologue of beta-L231, g2I242 is not displayed and neither are it‘s homologues. This reviewer feels that the presentation of the amino acids and the structural relationships could be greatly improved for more clarity.

Apparently RNA ratio was always 1:1:5, without any specific control experiments for the ratio of gamma- containing versus gamma-less pentamers. The specific “signatures“ of the mutated g2 subunits can be interpreted as confirmation of their presence, however, the presence of a1b2 pentamers cannot be ruled out and may lead to artefacts concerning the modulation by pTFD. Each expressed combination of WT and mutated constructs should thus also be characterized e.g. with a Zn2+ solution or with a benzodiazepine in order to gage the net integration of the (mutated) gamma2 subunits. Importantly: This reviewer considers the study worth of publication only if the gamma2 mutants are investigated concerning their assembly properties.

The “pCMBS exposure“ is defined in the methods section, but leaves the reader unable to replicate the findings as only the number is provided (e.g. in table 1), ranges from 30 to 9000 (quite a range!) and the actual values that were used for a given recording are not reported. Methods should be sufficiently complete to allow replication. Beside, the reasoning behind using this measure for “exposure“ should be explained, and the large range of conditions also should be explained and discussed. The reader should not have to search for a more detailed explanation in previous papers by the group.

Results including Figures and Supplementary Figures:

Figure S1 displays GABA CR curves for the 8 cysteine mutants. EC50 values are reported in table 1. The curves display quite different Hill slopes, which are not discussed at all. This has implications for the data obtained with sub-maximal [GABA], particularly the steep curve for g2L246C likely results in high variability. It is also not clear whether the subunit combinations result in receptors as drawn in Figure 1, or in a mixture of gamma2- containing and gamma-less pentamers.

Figure 2 and legend for Figure 2: The Figure depicts traces and provides Low/High rations. These are not the same ones as the numbers provided in the Figure legend. The reader is left puzzling, and this reviewer wants the legend to help graps the Figure including the numbers without the need to do computations. Lines 239/240: “The post-modification Low/High GABA response ratio in the control example (Fig. 2A) was 0.12, resulting in a modification ratio of 4.0 for this oocyte.“ The methods section is also not clear on this. The formulae could be provided in the Figure legend to help the reader.

Figure 3 and the legend for Figure 3: This is the core Figure, summarizing the results from all successful SCAMP datasets. Maybe this reviewer missed it – but the Figure legend fails to explain why MPAB (green bars) was not tested for two of the three gamma mutants. This reviewer appreciates the depiction of individual data points. It is interesting to note that some constructs appear to behave much more self- consistent than others. Maybe checking for consistent incorporation of gamma2 in all subunits combinations could explain some of the more variable datasets. In any case, some discussion seems in order. Panel F: has an outlier test been performed for the black data point at ~14? Panel F reports results for g2I242C, which is not depicted in Figure 1 at all. For all panels except G, the legend informs on + or -.

Discussion:

From the two gamma2-minus positions, one is “protected“, the other is not. To make the argument for a binding site more convincing, the pocket geometry should be considered and depicted (as part of the Discussion chapter perhaps).

Table 2 is provided to give an overview of previously published data and the submitted study, in an attempt to consolidate findings from several papers. The table is hard enough to grasp, and the legend is truly inacceptable. The numbers in the table seem to mean references 9, 12, 13 and 15 – not hard to figure, but the table legend really should provide all information that is needed to interpret the table without guesswork or extra effors. The + symbols presumably stand for “positive finding“, i.e. changed function, and the other symbols (minus and underscores) indicate “no effect“ – or something like that. The authors should explain. The mutants are not specified – are all twelve listed amino acids mutated to cysteine? The reader wants to know, without having to look up references 9, 12, 13 and 15.

A figure that displays all the amino acids in table 2 as part of the Introduction would be hugely helpful alongside with an explanation for the reader what was already known, and how the submitted study was designed.

The authors claim that they de-orphanized the alpha1+/ gamma2- pocket, and that the site at which “azi-ISO“ binds is different, closer to the extracellular domain. Indeed, the amino acids a1P278, a1S276 are further towards the extracellular domain, but g2Y241 is very close tot he region studied in the submitted work. How can the authors exclude the presence of a large “cleft“ which can form binding sites in either place, depending on the induced fit state? The claim that the described site for pTFD is novel and unique should be toned back, and the Discussion should correctly reflect the close proximity of the two proposed sites, or subsites.

In lines 330-335, the authors suggest that more selective compounds could be developed, which is plausible – but reference 22 (dealing with extracellular domain binding sites) is inappropriate to make the point. In the pockets discussed in the submitted study, most subunits are conserved within their class and thus only a few isofomrs can be targeted selectively. A simple alignment image in the supplementary information would be welcome, and a precise statement which subtypes could be targeted selectively.

Quality of writing:

The work gives the impression that it was hastily written with little or no quality control. For example, EC values (such as EC3, EC5) are written with or without subnscript, EC50S is used but not defined, table and figure legends do not fully describe the contents of the items, table 2 appears to contain a mix of minus (-) and underscore (_) symbols which are not defined at all. The authors claim (line 71) that pentobarbital was studied in reference 3 – this is not correct, the Kim et al. Cryo-EM study features a structure with phenobarbital.

**Do you want your identity to be public for this peer review?** For information about this choice, including consent withdrawal, please see our Privacy Policy

Reviewer #1: No

Reviewer #2: No

---

## [Author Response · Author response to Decision Letter 1]

29 Sep 2025

Response: We have formatted the manuscript, headers, citations, and references in compliance with the information provided. Figures and supporting materials were processed using PACE and named in accordance with instructions.

We combined figures and legends in supporting materials as pdfs. We did not include the legends in the manuscript text. Please let us know if a different approach to our supporting material is required.

“SAF received supported for this work from a grant from the US National Institutes of Health (R35GM141951) and the Dept. of Anesthesia Critical Care & Pain Medicine, Massachusetts General Hospital, Boston, MA, USA.”

Response: We have added a statement indicating that funders had no role (lines 369-370).

Response: Not applicable. However, we did add two new references (#’s 17 and 19) addressing the zinc sensitivity and receptor assembly issues.

5. Review Comments to the Author

Reviewer #1: The manuscript "Substituted cysteine modification and protection indicates selective interactions of the anesthetic photolabel pTFD-di-iPr-BnOH with α+/β- and α+/γ- transmembrane subunit interfaces of synaptic GABAA receptors" by Bhave and Forman investigates the interfaces between adjacent subunits of GABA-A receptors at which the anesthetic photolabel pTFD-d-iPr-BnOH binds, and compares those results to those found with two other agents, etomidate and R-mTFD-MPAB. All of these compounds, which act as positive allosteric modulators of GABA-A receptors, are shown to possess unique inter-subunit binding sites. The studies appear to have been painstakingly carried out and the manuscript is written in a clear and straightforward manner, although I do have a suggestion that might improve clarity.

I have no major concerns about this study and the conclusions reached, and my comments below are primarily suggestions that might improve it.

Response: Thank you for the positive overall assessment of this work.

1) What happens to the degree of modulation produced by these agents if one omits the gamma2 subunit, only expressing alpha/beta heteromeric receptors? Presumably etomidate modulation would not change much but what happens to MPAB and BnOH modulation if the gamma subunit contribution to binding is removed? Would one be able to obtain information about the relative importance of the two intersubunit binding sites of MPAB and BnOH by comparing their responses on alpha1beta3gamma2 vs alpha1beta3 receptors? Data provided in the paper already suggest that this might be so. In Figure 3, cysteine mutants at the alpha (+) / beta (-) interface (Fig 3B-D) show marked effects of MPAB in blocking pCMBS binding, but the effect is considerably smaller at the gamma (+) / beta (-) interface (Fig 3H). In contrast, it looks like the gamma2 subunit plays a greater role in the effects of BnOH, since mutations at alpha1 A291C and gamma2 L246C seem to be similar in terms of BnOH protection against pCMBS binding.

Response: This is an interesting suggestion that goes far beyond our experimental aims. However, expression of subunit mixtures lacking γ2 results in both 2α:3β and 3α:2β receptors in uncertain ratios. Thus, we would not expect the suggested experiment to provide a clear test of the interfacial drug binding effects. We also do not feel that interpretation of the pCMBS modification results at one anesthetic concentration are an accurate indicator of relative affinity for different interfacial sites. To draw such a conclusion would require concentration-dependence studies (a great deal of additional experimental work). Additionally, the sites are mutated in our experiments which might alter affinities for anesthetics. The case of βN265C, which we discuss, is a dramatic example.

2) Why was MPAB not tested in Figs 3F (gamma2 I242C) and 3G (alpha1 L246C), if only for the sake of completeness?

Response: This issue is also a concern of Reviewer #2, and we have added text to clarify our experimental strategy. Each of these protection studies represents a large amount of experimental work and the focus of our study was pTFD-di-iPr-BnOH and its possible interaction with the α+/γ– interface. We studied two cysteine mutants on each side of that interface, using etomidate as a single negative control drug for the two γ– mutants. Etomidate was chosen in these cases because its selectivity for β+/α– binding sites is best established by multiple approaches. For the two α+ mutants, R-mTFD-MPAB served as a positive control, while etomidate served as a negative control.

3) The location of the gamma2 I242 residue is not depicted in Fig. 1 and no data in Fig 3 is provided for the gamma2 S280 mutation to cysteine that is shown in the gamma subunit in Fig 1. This should be made consistent, or at least explained.

Response: Thanks for noting these details. Fig 1 now depicts homologs of the four shown residues where ETO interacts (2 “+”-face and 2 “–“ face) in β+/α– interfaces (12 residues in all). We have edited the text (lines 93-97) to note that not all of the depicted residues were studied.

4) In the Discussion it might be worth highlighting that, even though both MPAB and BnOH act at the alpha (+) / beta (-) interface, they do so differently. In Fig 3 it is shown that there is a difference between MPAB and BnOH in their effects on pCBMS enhancement in the alpha1S270C mutant (only MPAB blocked), compared to the alpha1A291C and beta3L231C mutants (both blocked), so the binding of these agents at this interface may overlap but is not identical at this site. This is depicted in Table 2, but may still be worth highlighting in the Discussion.

Response: Thank you for this suggestion. We have added text to the discussion to highlight the differential results for MPAB and pTFD-di-iPr-BnOH at this interface (lines 315-319).

5) Something for the authors to consider in a follow-up study: Does BnOH binding to the alpha(+)/gamma(-) site interfere with benzodiazepine actions of these receptors?

Response: Thank you for this interesting suggestion.

6) The authors should consider breaking up Figure 3 into three figures, with each new figure focused on alpha, beta or gamma subunit mutants. Each figure should also contain a small version of the graphic currently shown in Fig. 1, pointing out the mutations and subunit interfaces being dealt with in that figure. In reviewing this manuscript, I was constantly going back and forth between Figs. 1 and 3, trying to keep things straight. Maybe the current Fig 1. could then become the last figure in the paper, that summarizes the findings.

Response: We appreciate the difficulty readers may have in interpreting Fig 3 and we like this suggestion. We have split Fig 3 into 3 separate figures, one for each subunit, with diagrams indicating the interfacial sites probed by each mutation.

Minor typo: line 283, "findinds"

Response: Thank you. This typo has been corrected.

Reviewer #2: Bhave and Forman present a study concerned with the binding sites of a photoreactive compound „pTFD-di-iPr-BnOH“, in this review called pTFD for short. They use a cysteine based approach called SCAMP, and conclude that pTFD interacts with two sites at a1b2g2 receptors – namely the a1+/b2- and a1+/g2- binding sites in the TMD that are homologous with the „etomidate- site“ at b2+/a1-. The experimental approach has been used in the past and is thus well established. The results are of interest. The reasoning behind the study design, details in the methods, and the presentation and interpretation of the results can and should be improved for more clarity as indicated point by point below. Most suggested changes concern data analysis and presentation of the structural aspects and the results, but some additional control experiments are also needed.

Introduction:

1) Figure 1 is meant to support the Introduction and to provide information about the localization of the amino acids that were mutated to Cys in this and previous work. It depicts a schematic pentamer with the position of amino acids involved in the binding sites of interest. This schematic view is somewhat misleading, as it suggests that all three amino acids that line a given binding site are in the same plane of the TMD – which is actually not the case. For unknown reasons, the authors selected a M1 position which is further towards the intracellular end compared to those in M1 and M3. A rendering of an available experimental structure to show the positions of the mutated amino acids would be more informative compared to the presented scheme. If the authors prefer the schematic rendering, they should clearly state that the amino acids are in different planes relative to the ligand position. Moreover, the reader would be helped by a rendering with all 12 amino acids that are displayed in table 2.

Response: We have amended the Fig 1 legend to state that the labeled amino acids are located at varying depths along the extracellular to intracellular axis. We have also modified the figure to include all 12 residues listed in Table 2.

2) The explanation in lines 51/52 on the „+“ and „-„ sides of the binding sites is lacking M2, which contributes to both. A more complete legend for Figure 1 would solve this problem.

Legend for Fig 1: What is a „subsite“, what are subsite selective compounds???

Response: We have amended the text to state that aspects of M2 helices contribute to + and – interfaces (lines 51-52). We changed ‘subsite’ to ‘site’.

3) Methods and study design:

The reasoning behind the selection of the amino acids is not provided at all. See also the remark concerning Figure 1, the choice of the amino acid on M1 (- side) is non- obvious. While the reader learns in the discussion why table 2 features 12 amino acids, figure 1 displays nine, and the results from 8 mutants (partly displayed in Fig 1) are discussed, the reader would be helped if a table provides clarity, or if table 1 is extended to reflect the whole situation that is also summarized in table 2. For gamma2 minus, two mutants were studied, which is reflected correctly in the methods and in the results chapter. While g2L246 (displayed in Fig 1) ist he homologue of beta-L231, g2I242 is not displayed and neither are it‘s homologues. This reviewer feels that the presentation of the amino acids and the structural relationships could be greatly improved for more clarity.

Response: We have revised Fig 1 and its legend, as noted above. We also added more text at the end of the Introduction explaining our approach in the selection of mutations (lines 93-97). Briefly, we did not aim for the complete set of 12. We chose one + and one – mutation for each subunit, based mostly on which mutants we had in our freezer, and in the case of the α+/γ– interface, we studied two mutations on each side.

4) Apparently RNA ratio was always 1:1:5, without any specific control experiments for the ratio of gamma- containing versus gamma-less pentamers. The specific „signatures“ of the mutated g2 subunits can be interpreted as confirmation of their presence, however, the presence of a1b2 pentamers cannot be ruled out and may lead to artefacts concerning the modulation by pTFD. Each expressed combination of WT and mutated constructs should thus also be characterized e.g. with a Zn2+ solution or with a benzodiazepine in order to gage the net integration of the (mutated) gamma2 subunits. Importantly: This reviewer considers the study worth of publication only if the gamma2 mutants are investigated concerning their assembly properties.

Response: Yes, our study design assumed that the receptors we studied possessed the stoichiometry and arrangement shown in Fig 1. There are several reasons why we are confident that the vast majority of our receptors incorporate γ subunits and are arranged as shown in Fig 1. First, for the γ mutants that the reviewer feels must be further studied, the irreversible functional effects of pCMBS modification can only reflect receptors containing the mutated subunits, because no functional changes are produced when no cysteine substitutions are present (thus, α/β receptor currents cannot contribute to the γ mutant SCAMP results). Additionally, pTFD modulation of the mutants was not a major aim. It was a prerequisite for the SCAMP studies. Secondly, our control experiments using etomidate and R-mTFD-MPAB indicate that the mutated receptor assemblies were arranged the same way that many prior studies have demonstrated, including cryo-EM. Thirdly, the 1:1:5 mRNA mixture ratio that we have used for many years was shown in 2002 by Baumann et al (PMID: 12324466; now Ref # 17) to assemble into receptors that had the same stoichiometry and arrangement as receptors formed from concatenated subunit assemblies that constrained subunit stoichiometry and arrangement into the now-canonical configuration shown in Fig 1. Finally, several decades of our research using this approach in wild-type and mutant GABAA receptors, including studies of benzodiazepine modulation, have been consistent with this structure.

Despite this, we also acknowledge that data is more persuasive. So, we performed Zn2+ inhibition studies in our wild- type α1β3γ2L receptors, which showed only 6% inhibition at 100 µM zinc. This result indicates that the vast majority of the receptors we studied incorporated γ subunits that confer zinc insensitivity.

We have modified the methods (lines 154-163) , results (lines 239-242), and discussion (lines 3097-318) of the paper to address this issue.

5) The „pCMBS exposure“ is defined in the methods section, but leaves the reader unable to replicate the findings as only the number is provided (e.g. in table 1), ranges from 30 to 9000 (quite a range!) and the actual values that were used for a given recording are not reported. Methods should be sufficiently complete to allow replication. Beside, the reasoning behind using this measure for „exposure“ should be explained, and the large range of conditions also should be explained and discussed. The reader should not have to search for a more detailed explanation in previous papers by the group.

Response: We have altered Table 1 to report both the pCMBS concentrations and the exposure times used in SCAMP experiments for each mutant.

We added more text explaining why we chose partial modification conditions for SCAMP experiments (lines 210-212). In essence, we balanced the reduced magnitude of functional modification signals obtained under initial modification rate conditions (low pCMBS) against the intrinsic challenge of testing non-covalent ligands competing with a covalent and irreversible interaction.

Results including Figures and Supplementary Figures:

6) Figure S1 displ

---

## [Decision Letter · Decision Letter 1]

28 Oct 2025

Substituted cysteine modification and protection indicates selective interactions of the anesthetic photolabel pTFD-di-iPr-BnOH with α+/β- and α+/γ- transmembrane subunit interfaces of synaptic GABAA receptors

PONE-D-25-38999R1

Dear Dr. Forman,

We’re pleased to inform you that your manuscript has been judged scientifically suitable for publication and will be formally accepted for publication once it meets all outstanding technical requirements.

Kind regards,

Uwe Rudolph

Academic Editor

PLOS ONE

Additional Editor Comments (optional):

Reviewers' comments:

Reviewer's Responses to Questions

**Comments to the Author**

Reviewer #1: All comments have been addressed

Reviewer #2: All comments have been addressed

2. Is the manuscript technically sound, and do the data support the conclusions?

Reviewer #1: Yes

Reviewer #2: Yes

3. Has the statistical analysis been performed appropriately and rigorously?

Reviewer #1: Yes

Reviewer #2: Yes

4. Have the authors made all data underlying the findings in their manuscript fully available?

Reviewer #1: Yes

Reviewer #2: Yes

5. Is the manuscript presented in an intelligible fashion and written in standard English?

Reviewer #1: Yes

Reviewer #2: Yes

Reviewer #1: The authors have fully responded to the suggestions that I made, to hopefully improve an already strong submission. I have no further questions and commend them on a very nice study.

Reviewer #2: All points were addressed. The reviewer still feels that the extremely steep Hill slopes should be discussed, but as the conclusions remain unaffected by changes in Hill slope doe to the mutations., this matter of opinion should not slow down publication.

**Do you want your identity to be public for this peer review?** For information about this choice, including consent withdrawal, please see our Privacy Policy

Reviewer #1: No

Reviewer #2: No

---

## [Editor Report · Acceptance letter]

PONE-D-25-38999R1

PLOS ONE

Dear Dr. Forman,

I'm pleased to inform you that your manuscript has been deemed suitable for publication in PLOS ONE. Congratulations! Your manuscript is now being handed over to our production team.

Kind regards,

on behalf of

Dr. Uwe Rudolph

Academic Editor

PLOS ONE